# Delayed Diagnosis and Misdiagnosis of Lacrimal Sac Tumors in Patients Presenting with Epiphora: Diagnosis, Treatment, and Outcomes

**DOI:** 10.3390/diagnostics14212401

**Published:** 2024-10-28

**Authors:** Yu-Chen Chu, Chieh-Chih Tsai

**Affiliations:** 1Department of Ophthalmology, Taipei Veterans General Hospital, Taipei 112201, Taiwan; davidchu0966@gmail.com; 2Department of Ophthalmology, School of Medicine, National Yang Ming Chiao Tung University, Taipei 112201, Taiwan

**Keywords:** epiphora, lacrimal sac tumor, dacryocystitis, hemolacria, lacrimal irrigation, probing

## Abstract

Background/Objectives: Epiphora, or excessive tearing, is a common symptom often attributed to benign conditions such as dry eye or nasolacrimal duct obstruction. However, it can also be an early indicator of lacrimal sac tumors, which are frequently misdiagnosed or diagnosed late due to their subtle presentation. This study aims to identify the clinical features that contribute to delays and misdiagnoses of lacrimal sac tumors in patients presenting with epiphora, with the goal of improving early detection and treatment outcomes. Methods: This retrospective study reviewed medical records from Taipei Veterans General Hospital between 2007 and 2023, focusing on patients who presented with epiphora and were later diagnosed with pathologically confirmed lacrimal sac tumors. Inclusion criteria were limited to cases that were initially misdiagnosed or had a delayed diagnosis, with imaging and clinical evaluations confirming tumor-related tear drainage obstruction. Patients with non-tumor causes of epiphora were excluded. Results: Eleven cases of lacrimal sac tumors were identified, including two benign and nine malignant tumors. The average duration from symptom onset to diagnosis was 22.4 months. Common symptoms included epiphora (100%), discharge (54.5%), and hemolacria (18.2%). Subtle clinical signs, such as asymmetry in the medial canthal region and non-tender swelling, were frequently noted. Despite receiving appropriate surgical and adjuvant treatments, the impact of delayed diagnosis was significant. Two patients succumbed to tumor-related disease; one developed lung metastasis 12 years after diagnosis, and another experienced recurrence during a six-year follow-up after undergoing extensive exenteration, adjuvant chemotherapy, and radiotherapy. Conclusions: Lacrimal sac tumors can present insidiously with symptoms often mistaken for benign conditions, leading to significant diagnostic delays. Thorough history taking, meticulous physical examination, and timely imaging are crucial for early detection. Increased clinician awareness and a high index of suspicion for lacrimal sac tumors in patients with atypical epiphora are essential to improve prognosis and reduce the risk of severe outcomes.

## 1. Introduction

Epiphora, or excessive tearing that spills onto the cheek, is a common condition encountered by ophthalmologists and primary care physicians. It typically results from two main causes: tear overproduction and obstruction of tear drainage. Tear overproduction often occurs in response to ocular irritation or conditions such as dry eye, poor tear film, or infrequent blinking. Obstruction of tear drainage can occur at various anatomical sites, including the punctum, canaliculus, lacrimal sac, and nasolacrimal duct, or due to eyelid issues like misalignment or laxity [1].

While most cases of epiphora are due to common causes like anatomical abnormalities in the tear ducts, dry eye with reflex tearing, or other multifactorial factors [2,3,4,5], it is crucial for clinicians to recognize that epiphora can also be an early sign of lacrimal sac tumors [6]. The lacrimal drainage system’s diverse epithelial linings contribute to the development of both benign and malignant tumors. Among benign tumors, papillomas are the most frequent (36%), particularly the inverted type, which often recurs and has the potential to transform into malignant forms [7,8,9]. Squamous cell carcinoma (SCC) is the most common malignant tumor in this system, alongside other types such as transitional cell carcinoma (TCC), mucoepidermoid carcinoma (MEC), adenocarcinoma, adenoid cystic carcinoma (ACC), and lymphoepithelial carcinoma [10]. Non-epithelial tumors, accounting for about 25% of lacrimal sac tumors, may arise from disruptions in lacrimal drainage-associated lymphoid tissue, Epstein–Barr virus infection, or immunodeficiency [11,12,13,14]. Diffuse large B-cell lymphoma (DLBCL) is the most common primary sac lymphoma (43%), followed by MALT lymphoma (24%) and unclassified B-cell lymphoma (21%) [15].

Though rare, these tumors can have severe, potentially life-threatening consequences if not promptly diagnosed and treated. Suspicious signs of malignancy include a firm, immobile mass near the medial canthal tendon, epistaxis, and proptosis [7,16,17,18,19,20,21,22,23]. Unfortunately, the insidious nature of many lacrimal tumors, which often present with nonspecific symptoms like epiphora alone, can lead to delays in diagnosis and misdiagnosis as more benign conditions such as lacrimal duct obstruction or dacryocystitis [11,16,24,25,26,27,28].

This article aims to highlight the clinical features that can help reduce diagnostic delays and misdiagnoses of lacrimal sac tumors in patients presenting with epiphora. By illustrating clinical presentations, treatment approaches, and the severe outcomes seen in cases of delayed or misdiagnosed tumors, we aim to enhance clinician awareness and improve early detection and management, ultimately reducing the risk of adverse outcomes for these patients.

## 2. Materials and Methods

In this retrospective study, we analyzed medical records from Taipei Veterans General Hospital, focusing on patients who presented with a chief complaint of epiphora and were subsequently diagnosed with pathologically confirmed lacrimal sac tumors between 2007 and 2023. The inclusion criteria were specific to patients who were initially misdiagnosed or experienced delays in the diagnosis of lacrimal sac tumors, with epiphora as the primary presenting symptom.

The exclusion criteria were meticulously crafted to ensure the study concentrated exclusively on tumor-related epiphora. Patients with epiphora resulting from tear overproduction due to non-tumor factors, such as ocular surface diseases, allergic conjunctivitis, or dry eye syndrome, were excluded, as these conditions do not involve structural obstruction by a mass. We also excluded cases of lacrimal drainage obstruction not caused by lacrimal sac tumors, such as acute dacryocystitis, where inflammation rather than a neoplasm caused blockage. Additionally, patients were excluded if imaging revealed obstruction confined to the nasolacrimal duct without involvement of the lacrimal sac, such as isolated nasolacrimal duct stenosis, to prevent confounding from unrelated pathologies.

A comprehensive ocular history was obtained for each patient, including details on epiphora, medication use, history of dacryocystitis, allergies, as well as the frequency and duration of symptoms. This was followed by a detailed slit-lamp examination, assessing tear meniscus height, tear film break-up time, and results of the fluorescein dye retention test to identify reflex tearing associated with dry eye syndrome. Additionally, we evaluated for eyelid margin diseases such as meibomian gland dysfunction, blepharitis, and trichiasis and assessed for eyelid malposition or paralysis. Physical examinations commenced with a thorough inspection of the bilateral canthal regions, focusing on symmetry and the concavity of the medial canthal areas (Figure 1). While mild asymmetry may be observed, any fullness or mass above the medial canthal tendon raises a high suspicion for malignancy. Palpation is essential to assess for subtle fullness. Signs of pain, reflux discharge, erythema, and swelling beneath the medial canthal ligament tendon are indicative of acute dacryocystitis. If fullness or a mass-like lesion is detected during the physical examination, further imaging studies should be conducted.

Imaging played a crucial role in further evaluation, particularly when asymmetry in the bilateral canthal regions or a palpable mass was detected. Fourteen patients with suspected lacrimal tumors underwent computed tomography (CT), with or without magnetic resonance imaging (MRI), following thorough inspection and palpation. Among these, three patients showed only nasolacrimal duct obstruction, leading to distension of the lacrimal sac, and were excluded from further analysis. The remaining 11 patients were confirmed to have lacrimal sac tumors through surgical pathology. Following diagnosis, appropriate treatments were administered, and all patient outcomes were carefully recorded during follow-up.

This retrospective study was conducted in accordance with the Declaration of Helsinki and approved by the Ethics Committee of Taipei Veterans General Hospital, with approval code 2006-06-011CC and an approval date of 2023.

## 3. Results

A total of 11 histopathologically confirmed lacrimal sac tumors in patients initially presenting with epiphora were included in our study, with 2 (18.2%) benign tumors and 9 (81.8%) malignant tumors. Patient information is summarized in Table 1. The mean age is 56 (ranges from 18 to 82), with three males and eight females. Six patients (54.5%) had a history of presumed nasolacrimal duct obstruction, having undergone multiple probing or irrigation procedures, while one patient with lacrimal sac melanoma had previously undergone balloon dacryocystoplasty. The remaining five patients (45.5%) were previously treated for reflex tearing. Patients presented with various symptoms, with epiphora being the most commonly observed in all cases, followed by discharge (54.5%), hemolacria (18.2%), pain (9.1%), proptosis (9.1%), and epistaxis (9.1%).

Physical examination showed a non-tender mass in seven cases (63.6%), non-tender swelling in three cases (27.3%), and tender swelling in one case (9.1%). Seven lesions (63.6%) were located above the medial canthal tendon; some were obvious, while others were less apparent. The remaining four cases (36.4%) exhibited only mild asymmetry, necessitating palpation to raise suspicion of malignancy. The average duration from the onset of symptoms to the time of diagnosis was 22.4 months, with a median of 10 months and a range spanning from 2 to 120 months.

Two (case numbers 5 and 11) patients passed away due to tumor-related disease, one with ACC four years later and another with melanoma two years later. Additionally, one case (case number 3) of SCC developed lung metastasis 12 years after diagnosis. Despite receiving chemotherapy, targeted therapy, and multiple wide excisions of tumor margins, one patient (case number 6) with ACC experienced recurrence during the six-year follow-up period.

We highlight three patients’ clinical performances and outcomes as follows to demonstrate the importance of physical examination.

A 44-year-old man (case 3) had a history of recurrent caruncle papilloma in his left eye, which had been excised and treated with cryotherapy more than ten times since he was 20 years old. He presented to our outpatient department with a six-month history of persistent epiphora and recurrence of a left caruncle mass with intermittent bleeding (Figure 2A). The patient also reported recent episodes of epistaxis. Despite seeking treatment at a local clinic, his symptoms persisted, prompting him to seek further evaluation at our facility. Slight swelling was observed on the left medial canthus upon comparing bilateral symmetry, and non-tender swelling was noted during palpation. Slit-lamp examination showed a papilloma-like mass over the left caruncle (Figure 2B). A CT scan was ordered, revealing a lobulated, heterogeneous, enhancing soft tissue nodular lesion within the left lacrimal sac associated with bony destruction and expansion of the nasolacrimal duct (Figure 2C). A dacryocystectomy with adjunct MMC-C soaking was performed along with the removal of the left maxillary sinus medial wall, the anterior and posterior ethmoid sinus, the middle and inferior turbinate, and the left parietal anterior skull base (Figure 2D). Pathological examination confirmed the presence of SCC. Following the surgery, concurrent chemoradiotherapy was administered. However, twelve years later, the patient developed lung metastasis.

A 63-year-old woman (case 6) presented to medical professionals with complaints of epiphora in her left eye, which was more pronounced than in her right eye. Six months prior, she had visited a local clinic where probing and irrigation were conducted, diagnosing a nasolacrimal duct obstruction. Upon inspection at our clinic, slight asymmetry was observed at the bilateral medial canthal region; however, during palpation, a non-tender mass was detected (Figure 3A). Subsequent CT scans confirmed a left lacrimal tumor with nasolacrimal duct involvement (Figure 3B). She underwent endoscopic medial maxillectomy and dacryocystectomy, with a pathological examination revealing an ACC. To ensure optimal surgical margins, exenteration and skull base surgery (Figure 3C) were performed via the endoscopic endonasal approach (EEA) one week later. Additionally, she underwent chemotherapy and radiation therapy. However, six years later, a recurrence of ACC and radiation-induced sarcoma (Figure 3D) was observed in the posterosuperior part of the left orbital fossa near the skull base, necessitating extensive resection followed by combined immunotherapy (Pembrolizumab) and radiotherapy.

A 57-year-old woman (case 11) had been enduring persistent epiphora in both eyes for over two years. She sought medical attention at a local clinic, where she underwent multiple probing procedures. Two months preceding her visit to our outpatient department, she underwent bilateral balloon dacryocystoplasty. Upon examination at our clinic, mild swelling was noted over the left medial canthal region, and hemolacria was observed following lacrimal irrigation and probing (Figure 4A). A CT scan revealed a mass in the left lacrimal sac (Figure 4B). An incisional biopsy was performed, revealing a pigmented mass ultimately diagnosed as melanoma (Figure 4C). MRI of the left lacrimal lesion indicated high signal intensity at T1W and low signal intensity at T2W (Figure 4D,E). Subsequent to diagnosis, the patient underwent dacryocystectomy, canaliculectomy, and medial maxillectomy (Figure 4F). Postoperatively, she received adjuvant chemotherapy and radiotherapy. Despite these comprehensive interventions, the patient succumbed to metastasis in the lungs, bones, and brain two years later.

In each of the three cases presenting with epiphora, subtle swelling was noticeable around the medial canthal region (Figure 2A, Figure 3A, and Figure 4A). Through meticulous inspection and palpation, potential lacrimal sac lesions could be detected, prompting the arrangement of imaging studies to promptly confirm the presence of a lacrimal sac tumor.

## 4. Discussion

Epiphora refers to the overflow of tears, which can result from either excessive production by the lacrimal gland or obstruction within the lacrimal drainage system. Anatomical abnormalities in the lacrimal passage and oversecretion of tears, as seen in ker-atoconjunctivitis, allergies, or dry eye with secondary reflex tearing, have been reported as the most common causes of epiphora in previous studies [3,4,5,29,30,31,32,33]. Lacrimal sac swelling and/or a mass may be due to long-term nasolacrimal duct obstruction associated with inflammation of the lacrimal sac. While tumor development is rare, it is crucial to consider that lacrimal sac tumors can also cause obstruction of tear drainage. Due to a complex anatomy and a concealed location, the clinical appearance of lacrimal sac neoplasms in the initial stages can be deceptive, often manifesting solely as simple epiphora with a common cause before a visible mass is identified [11,16,34]. In the present study, the average duration from the onset of symptoms to the time of diagnosis was 22.4 months, with a median of 10 months and a range from 2 to 120 months.

Despite the challenge of diagnosing these elusive tumors, we present our experience in managing patients with lacrimal sac tumors that were previously misdiagnosed. Thorough history taking and a comprehensive physical examination are essential in such cases. The concavity of the bilateral medial canthi should be carefully compared (Figure 2A, Figure 3A, Figure 4A, and Figure 5). The asymmetry and a less prominent tear trough suggest a possible lesion at the lacrimal sac. Palpation should always be performed after careful inspection. Swelling and a mass may be present due to a tumor filling the sac and causing obstruction. If abnormalities are detected during the physical examination, further imaging studies should be conducted. CT scans can enable the detection of a lacrimal sac mass and assessments of any bony erosion or invasion into surrounding structures [7,18,20,22]. For clear visualization of adjacent soft tissues and differentiation between cystic or inflammatory masses and solid tumors, MRI with both T1- and T2-weighted sequences is recommended. Lesions typically appear isointense on T1-weighted and T2-weighted sequences, with restricted diffusion observed on diffusion-weighted images [35,36].

The most effective management approach for lacrimal sac tumors depends significantly on factors such as histological type, tumor size, and the patient’s overall health. More than 55% of these tumors are malignant, with the malignancy rate ranging from 55% to 100%. These tumors are generally locally invasive and are associated with a high recurrence rate [11]. Surgery is typically the primary treatment option, often followed by postoperative radiotherapy or adjuvant chemotherapy [9,36,37]. Due to the anatomical location of the lacrimal sac and its connections, tumors can grow internally, extending towards the orbit, nasal cavity, sinuses, and skull base. In cases requiring surgical intervention, locally advanced tumors may necessitate the removal of adjacent structures such as the eyeball, nasal bone, maxillary bone, or sinuses [10,16,34]. For instance, in case 6, ACC invading the skull base required exenteration and multiple wide resections. Despite treatment, carcinoma recurrence rates approximate 50%, with mortality rates ranging from approximately 37% to 100% and increasing significantly with recurrence [38]. These tumors also exhibit a propensity for metastasis and high fatality rates [16,17,36,39,40,41], as seen in case 3 with SCC leading to lung metastasis, and case 11 with melanoma causing widespread systemic metastasis, ultimately resulting in death.

Given the rapid advancements in ophthalmic instruments, emphasizing the importance of thorough clinical examinations is paramount. Epiphora, encountered frequently by ophthalmologists, often necessitates evaluating the lacrimal drainage system through procedures like lacrimal irrigation, probing, or dacryocystoplasty. However, obstruction due to lacrimal sac tumors, though less common, is a significant concern. These tumors have the potential to spread to the orbit or nose, complicating early detection and management. In many instances, by the time tumors become clinically evident, they have already extensively infiltrated surrounding structures [11]. In our study, ACC and melanoma initially presented insidiously, causing only mild asymmetry (Table 1). This often led to diagnostic difficulties, with these tumors frequently being treated as nasolacrimal duct obstructions. Therefore, carefully following the steps of the physical examination illustrated in Figure 1 is crucial. Furthermore, arranging inappropriate tests without meticulous physical examination may delay diagnosis, as demonstrated by the patients in the present study (Table 1), many of whom underwent procedures like lacrimal irrigation or probing beforehand. This approach can also pose risks, as exemplified in case 11 with lacrimal sac melanoma, where dacryocystoplasty may have inadvertently promoted metastasis, thereby endangering the patient’s life.

This retrospective study on lacrimal sac tumors is limited by a small sample size of only 11 patients, which may not provide a comprehensive representation of the disease spectrum. Additionally, the variability in pathological diagnoses among the cases adds further complexity, potentially affecting the generalizability of the findings.

## 5. Conclusions

This study underscores the critical need for heightened clinician awareness of lacrimal sac tumors in patients presenting with epiphora, which are frequently misdiagnosed or diagnosed late due to their subtle early symptoms. Although rare, these tumors can have severe, life-threatening consequences if not promptly identified and treated. Our findings reveal an average delay of 22.4 months from symptom onset to diagnosis, highlighting the challenges in recognizing these malignancies. This study emphasizes the importance of thorough history taking, detailed physical examination, and appropriate imaging in patients with persistent or atypical epiphora. Subtle signs such as asymmetry in the medial canthal region or non-tender swelling may indicate an underlying malignancy. Early detection through comprehensive clinical evaluation and timely imaging is crucial for improving patient outcomes and reducing the risk of severe complications, including metastasis and recurrence. Given the high malignancy rates and metastatic potential of these tumors, ophthalmologists must maintain a high index of suspicion in patients with persistent or unusual epiphora to reduce diagnostic delays and enhance prognosis.

## Figures and Tables

**Figure 1 diagnostics-14-02401-f001:**
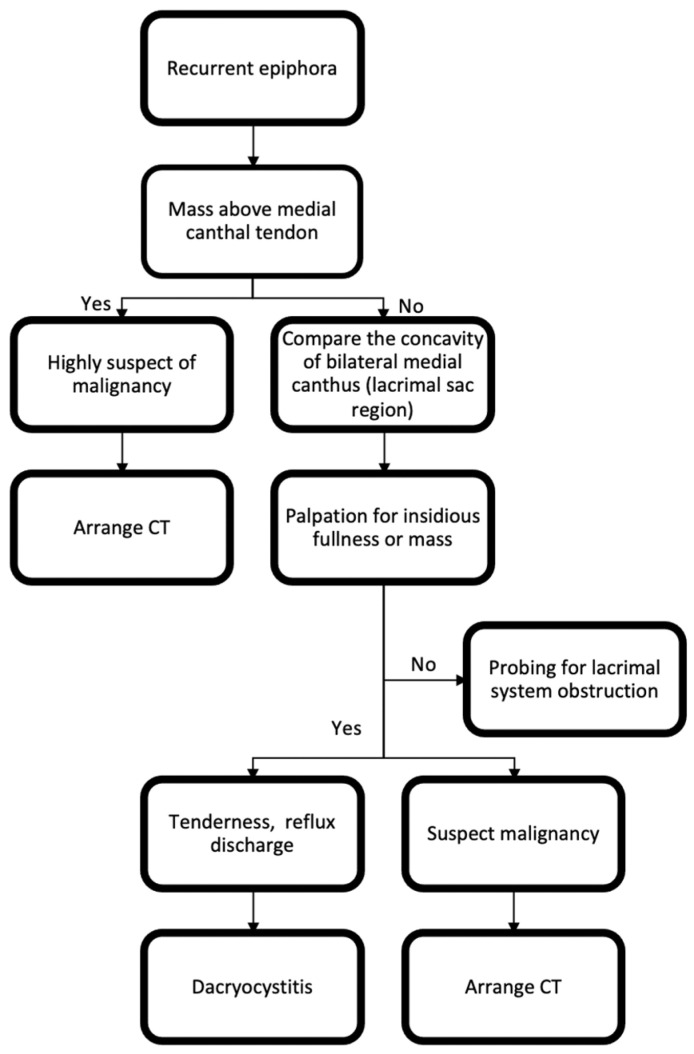
Steps for physical examination of potential lacrimal sac tumors.

**Figure 2 diagnostics-14-02401-f002:**
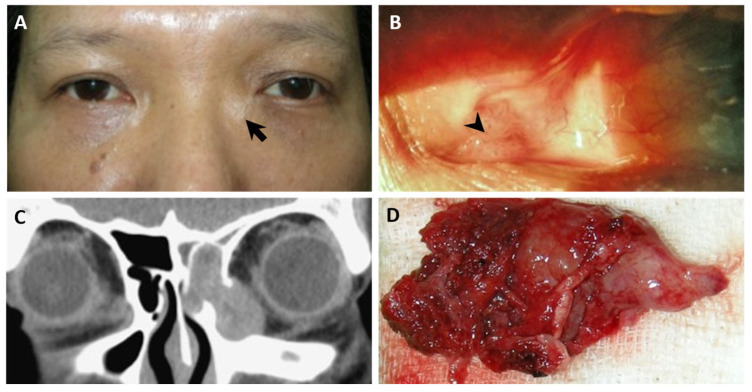
SCC of the lacrimal sac. (**A**) A 44-year-old male presented with a 6-month history of epiphora and recurrent caruncle papilloma with non-tender swelling noted at the medial canthal area (arrow). (**B**) A papilloma-like mass (arrowhead) over the left caruncle under the slit lamp. (**C**) The coronal CT scan displayed a non-calcified, soft-tissue, space-occupying mass in the region of the left lacrimal sac. (**D**) Gross specimen of SCC.

**Figure 3 diagnostics-14-02401-f003:**
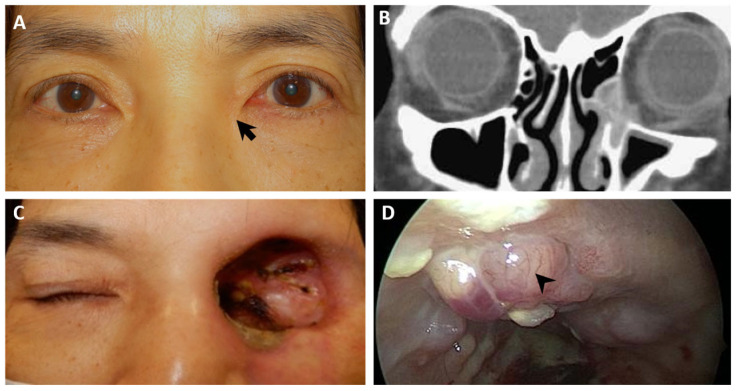
ACC of the lacrimal sac. (**A**) Six months of persistent epiphora following multiple probing and irrigation procedures in a 63-year-old female, with slight asymmetry (arrow) noted at the bilateral medial canthal region. (**B**) The coronal CT scan shows a soft-tissue, space-occupying lesion in the left lacrimal sac area, accompanied by widening of the bony nasolacrimal duct and minimal bone erosion. (**C**) The patient underwent left orbital exenteration and skull base surgery to achieve optimal surgical margins. (**D**) Six years after surgery, recurrent ACC (arrowhead) was observed in the posterosuperior part of the left orbital fossa during endoscopy.

**Figure 4 diagnostics-14-02401-f004:**
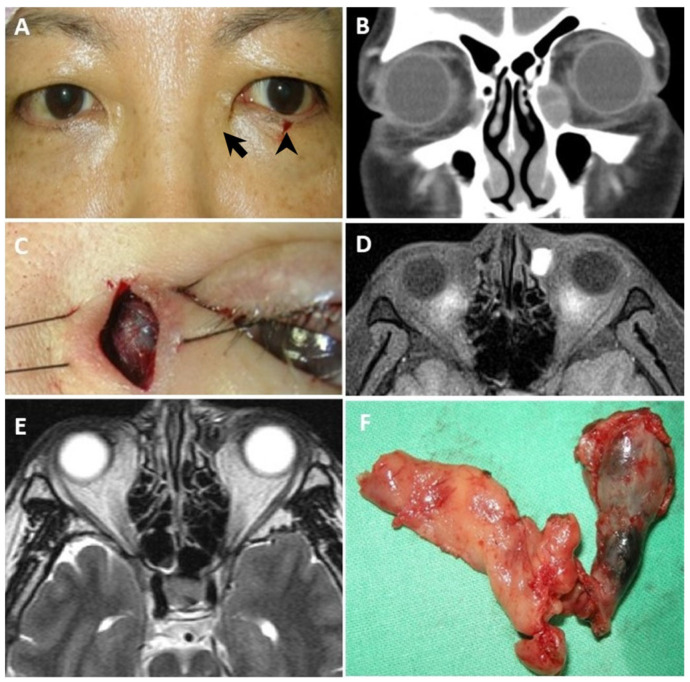
Malignant melanoma of the lacrimal sac. (**A**) After lacrimal irrigation and probing, hemolacria (arrowhead) was observed in a 57-year-old female with mild swelling of the left medial canthal region (arrow). (**B**) The coronal CT indicates a soft-tissue, space-occupying lesion located in the region of the left lacrimal sac. (**C**) During the incision biopsy, a darkly pigmented lacrimal mass was noted on the left side. (**D**) Left lacrimal sac lesion shows a high signal in T1-weighted MRI (axial view) and (**E**) a low signal in T2-weighted MRI (axial view). (**F**) The tumor extends from left lacrimal sac to nasolacrimal duct.

**Figure 5 diagnostics-14-02401-f005:**
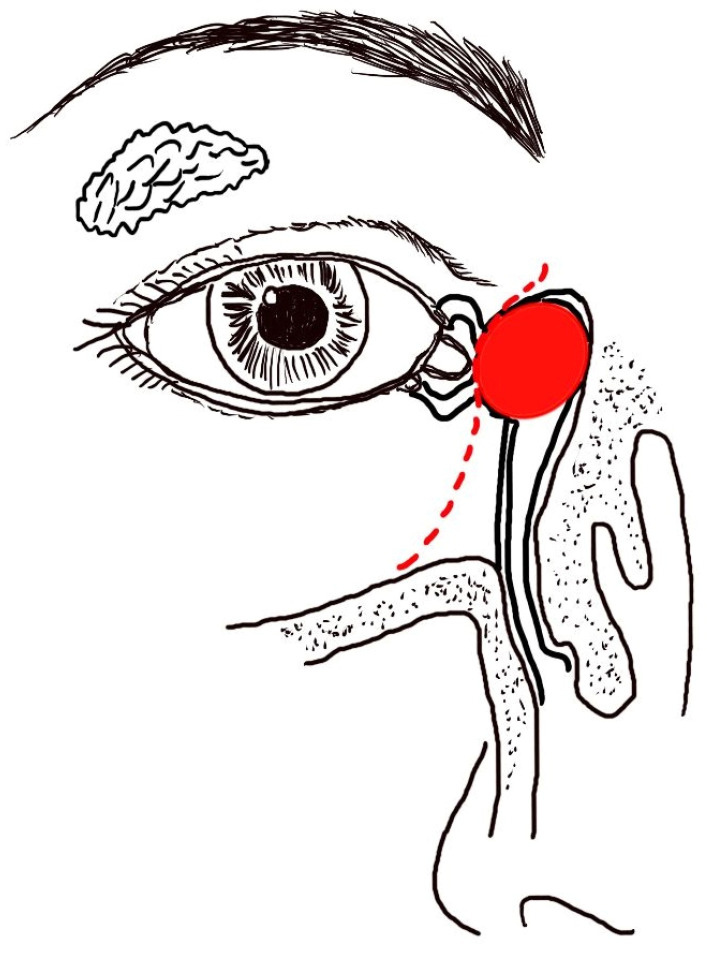
An illustration of a possible lacrimal sac lesion (red ellipsoid) shows changes in the concavity of the medial canthal region (dash line) that result in a less prominent tear trough.

**Table 1 diagnostics-14-02401-t001:** Baseline characteristics of 11 patients in this study.

Pt	Age Sex	Symptoms	Signs (PE)	Histology	Prior Procedure	Treatment	Time to Diagnosis (Months)	Outcomes
1	67M	Epiphora Discharge	Non-tender mass above MCT	Inverted papilloma	Lacrimal irrigation/probing (patent)	Surgery	120	No recurrence
2	18F	Epiphora	Non-tender mass above MCT	Solitary fibrous tumor	Lacrimal irrigation	Surgery	2	No recurrence
3	44M	Epiphora Hemolacria Epistaxis	Non-tender swelling above MCT	Squamous cell carcinoma	-	Surgery POCRT Immunotherapy	6	Lung metastasis
4	47F	Epiphora Discharge Pain	Tender Swelling Mild asymmetry	Squamous cell carcinoma	-	Surgery PORT	8	No recurrence
5	81F	Epiphora Discharge Itching	Non-tender swelling Mild asymmetry	Adenoid cystic carcinoma	Lacrimal irrigation	Surgery PORT	36	Expired
6	63F	Epiphora	Non-tender mass Mild asymmetry	Adenoid cystic carcinoma	Lacrimal irrigation/probing	Surgery POCRT Immunotherapy	6	Recurrence
7	45F	Epiphora	Non-tender mass above MCT	Diffuse large B-cell lymphoma	-	Surgery POCRT	17	No recurrence
8	55F	Epiphora Discharge	Non-tender mass above MCT	Diffuse large B-cell lymphoma	Lacrimal irrigation	Surgery POCRT	10	No recurrence
9	82M	Epiphora Proptosis	Non-tender mass above MCT	Lymphoepithelial carcinoma	-	Surgery PORT	14	No recurrence
10	57F	Epiphora Discharge	Non-tender mass above MCT	Follicular lymphoma	-	Surgery POCT	3	Lung metastasis
11	57F	Epiphora Discharge Hemolacria	Non-tender swelling Mild asymmetry	Malignant melanoma	Lacrimal irrigation/ probing/balloon dacryocystoplasty	Surgery POCRT	24	Expired

Abbreviation: MCT, medial canthal tendon; PORT, postoperative radiotherapy; POCRT, postoperative chemoradiotherapy; POCT, postoperative chemotherapy.

## Data Availability

The data presented in this study are available on request from the corresponding author.

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
