# Peer review of "Delayed Diagnosis and Misdiagnosis of Lacrimal Sac Tumors in Patients Presenting with Epiphora: Diagnosis, Treatment, and Outcomes"

_diagnostics, 2024, doi:10.3390/diagnostics14212401_

Round 1

Reviewer 1 Report

Comments and Suggestions for Authors

Manuscript is well written. Introduction provides sufficient background about the topic. Case presented here are well described and the figures are well represented. This study emphasizes the importance of early careful examination of the lacrimal sac tumors that is often misdiagnosed putting the patient life at risk. This is highlighted by the authors as an average delay of 22.4 months from onset to diagnosis.

In the method section authors should have not mentioned if they were required to take patients consent or exempted. Did they followed proper consenting guidelines as per their institutional/hospital approved protocol. 

It would be helpful if they can make a table to highlight the steps or features of lacrimal sac tumors with epiphora, that the clinicians should pay attention during physical examination. Authors should also address the limitations of this study. 

Comments on the Quality of English Language

English language is fine, minor spell check is recommended.

Reviewer 2 Report

Comments and Suggestions for Authors

Please, correct word e.g.Hemolcaria in "Key words" etc.

Give more details about patency of lacrimal dear ducts, probiung in your group of patients.

Reviewer 3 Report

Comments and Suggestions for Authors

Good paper. Please see the file attached for suggestions aimed at improving the manuscript.

Comments on the Quality of English Language

 Please proofread the manuscript for grammar, mainly for the tense used. 

Round 2

Reviewer 1 Report

Comments and Suggestions for Authors

Authors have addressed all the concerns and have improved the manuscript significantly.